# Aerosol Release by Healthy People during Speaking: Possible Contribution to the Transmission of SARS-CoV-2

**DOI:** 10.3390/ijerph17239088

**Published:** 2020-12-05

**Authors:** Thomas Eiche, Martin Kuster

**Affiliations:** 1Thomas Eiche GmbH, Gempenstrasse 50, CH-4133 Pratteln, Switzerland; 2Novartis Pharma AG, Novartis Business Services, CoE HSE, WSJ 503/13/50, 4002 Basel, Switzerland; martin.kuster@novartis.com

**Keywords:** SARS-CoV-2, COVID-19, aerosol transmission, breathing, speaking, asymptomatic spreader, meetings

## Abstract

Our research aimed to review the potential risk of infection by SARS-CoV-2. We used an excerpt of a data set generated in May 2020 for reviewing the SARS-CoV-2 prevention concept of orchestras, singers and actors. People were sampled for droplet release for one-hour activities using a Grimm spectrometer covering a spectrum of 1 to 32 µm diameter. We estimated the number of “quanta” in the exhaled liquid from viral concentrations of 10^6^ to 10^11^/mL, based on the Human Infective Dose 50 of 218 viral particles. We employed the Wells–Riley equation to estimate the risk of infection in typical meeting rooms for a one-hour meeting of 2, 4 and 6 people observing a 2 m distance. The four participating adults released a mean of 1.28 nLm^3^ while breathing, 1.68 nL/m^3^ while speaking normally, and two adults released a mean of 4.44 nL/m^3^ while talking with a raised voice. The combination of 50% breathing, 45% talking normally and 5% speaking with a raised voice increased the risk of infection above 5% for a one-hour meeting of two people. The result is based on 6 quanta released, corresponding to an initial virus concentration of 1000/nL (10^9^/mL) in the fluid of the upper respiratory tract. Our data confirm the importance of using facemasks in combination with other measures to prevent transmission of SARS-CoV-2 at the workplace.

## 1. Introduction

Unless showing symptoms of COVID-19, any person encountered might be potentially spreading SARS-CoV-2. Humans are naïve to this new disease entity. Epidemiological data, including the estimated reproduction number zero (R0) between 2 and 4, suggested early on the possibility of a combination of aerosol (inhalation) and droplet (contact) borne transmission of SARS-CoV-2 [1,2,3]. 

Vella and colleagues reviewed the modalities of transmission of SARS-CoV-2 in an extensive literature review. The virus is also detectable in considerable quantities in stool and urine. Therefore, contamination of surfaces contributing to the spread of COVID-19 may also occur via deposition of fomites [4].

Transmission of viruses most often occurs in indoor environments and in smaller clusters. Larger clusters, with many people infected during the time spent together, are documented in irregular intervals in churches, restaurants or bars, slaughterhouses, weddings, sport events or work (e.g., a call center) [5]. Transmission of SARS-CoV-2 may occur frequently by contact to pre-symptomatic and asymptomatic people [6]. In conclusion, droplet aerosols (diameter of ≤5 µm) may play a more important role in transmission of SARS-CoV-2 than initially anticipated.

Table 1 summarizes the main factors for transmission of SARS-CoV-2 and protection thereof, based in part on information from WHO [7].

In order to reduce the spread of SARS-CoV-2, many countries installed “lock downs”, effectively closing all non-essential businesses or activities. At the same time, businesses had the task of developing “protection concepts” before resuming their activities, once authorities allow easing of measures. Droplets as main mode of transmission need special attention in protection concepts.

Droplets form in the airways when air is moving over the liquid layers of mucus. Depending on the air-flow pattern during breathing, speaking, singing, coughing or sneezing, droplets of different sizes form at different levels of the airway system [8,9].

Bronchioli (most peripheral section of the bronchial tree): Here droplets of various sizes, starting at 1 µm or less will be produced when the shear stress of air movement over the fluid is strong enough. Coughing or labored breathing are necessary to reach the air velocities to form large amounts of droplets over a wide range of diameters. 

Larynx: Droplet formation is most efficient while speaking, yelling or singing. Droplets typically tend to have a diameter of more than 1 µm. 

The oral cavity: The oral cavity is the location, where predominantly large droplets of more than 100 µm diameter emanate. 

Droplets of <5 µm diameter (so called droplet aerosols or respirable aerosols) are capable of reaching the periphery of the lungs. Droplets larger than 10 µm may deposit in the upper respiratory tract (mucous membranes of nose, pharynx, throat, trachea) or, if between 5 and 10 µm (so called inhalable aerosols), in the middle to lower parts of the bronchial tree [10].

Several authors have published the numbers of droplets produced by breathing alone, while speaking, coughing or sneezing. The number and size of particles released per activity spans a substantial amount as different methods in sampling and counting of droplets have been developed and used over time [8,10,11,12,13]. 

Healthy subjects release slightly less particles in a larger spectrum of diameters, compared to people suffering from symptomatic infections of the upper and especially the lower respiratory tract [10].

Eminent scientists under the lead of Lidia Morawska and Don Milton have written a review to make the point of droplet aerosols playing an important part in disseminating SARS-CoV-2 [14].

In their view-point article, Klompas and colleagues summarized the evidence for an airborne transmission of SARS-CoV-2 as follows: “the balance of currently available evidence suggests that long-range aerosol-based transmission is not the dominant mode of SARS-CoV-2 transmission”, supporting the position of WHO [7,15].

Stadtnytskyi and colleagues estimated 1000 droplets in the aerosol fraction (<5 µm diameter) containing a single virus particle per minute speaking [11]. Using their count of 158,000 droplets per minute speaking (or 180 nL fluid per minute), a viral load of 7 × 10^6^/mL would yield a single virus in one out of 158 particles of the same size [11]. In other words; a single particle of less than 5 µm has a probability of 0.006 to contain a single virus, based on these calculations.

In this study, we contribute to the discussion by using aerosol release data of healthy volunteers over 60 min of breathing, speaking or speaking aloud. Based on the total amount of liquid released, we estimated an infection risk from aerosols in a one-hour meeting using increasing concentrations of virus. 

We deliberately focused on healthy volunteers, as these people reflect the physiology of asymptomatic people best. Asymptomatic people spread the virus unwittingly. 

However, their percentage in the general population may be substantial. Oran and Topol estimate the prevalence of asymptomatic COVID-19 infection to be 40% to 45% of those infected with SARS-CoV-2 [16].

## 2. Materials and Methods 

### 2.1. Study Population

The original data set with 48 participants in total is comprised of aerosol and droplet sampling data from wind musicians, actors, singers and a choir as part of the review of the protection concept of the Swiss National Organization for Orchestras and Stages. The time of the original sampling was from 9 May 2020 to 30 May 2020. The results of the work, including details concerning methods and the study population, are available on researchgate.net as a non-peer reviewed preprint [17].

Out of the original data-set, data for the full hour sample time are available for two actors (one female, aged 49 years, one male, aged 35 years) and two singers (one female, aged 30 years, one male, aged 62 years). 

The re-analysis of the data set was carried out following the rules of the Declaration of Helsinki of 1975. The project has been accepted by the Swiss Ethical Board Northwest (Reference number: 2020-02400; Decision date: 11 November 2020).

### 2.2. Sampling of Aerosols and Droplets

Sampling of the participants data used in this analysis took place on 14 May 2020. We sampled droplet concentrations as counts per liter using Grimm aerosol laser spectrometers, model 1.109 with laser light scattering, filter collector and 31 particle fractions with a diameter from 0.25 to 32 µm. The instruments were positioned in different setups for verifying the reliability of the method. The optimal set up proved to be 25 and 50 cm away from the mouth at 1.5 m height (Figure 1). The sampling interval of the devices is six seconds. A further TSI Dust Track II device was used for source location and background measurement (these data were not used for this work). All samplers were calibrated to have a detection limit of 0.1 µg/m^3^ dust (or 0.1 nL/m^3^ as liquid aerosols) and a volume flow of 1.2 L/min. Particles passing the measuring channel are trapped in an absolute filter to prevent repeat counting.

We measured absolute humidity and the CO_2_ at the sampling head in parallel, to demonstrate that the droplets sampled in the spectrometer come from exhaled air. Video recordings were used to assign the measurement data to the various activities (breathing, speaking, singing, yelling).

The measurements of the droplets by the four people used in our analysis took place on their regular theater stage with the audience empty and the ventilation running. Thus, accumulation of droplets in the room was excluded. Cleaning of the stage and audience controlled for background contamination with particles on levels as low as possible. The subjects behaved as if they would perform in front of an audience. Each participant presented texts of their choice or talked to the experimenter. 

We captured the sampling on video, too. Each six-second sampling sequence in the readout of the Grimm spectrometers was assigned to the activity recorded. The resulting pattern of the time slots formed the basis for our extrapolation to the full hour of breathing, speaking or speaking with a raised voice.

Figure 1 shows a schematic view of the applied sampling set up. The person speaks at a level into two aerosol sampling heads, 25 and 50 cm away from the mouth. CO_2_ and humidity levels are sampled, to verify exhaled air.

We used the concentration of droplets as counts per liter per channel from 1 to 32 µm diameter. Such a spectrum of droplets is or will be over time in the aerosol fraction of ≤5 µm diameter, due to shrinking.

Aerosol droplets of a diameter of <1 µm at the sampling point have a very low probability of containing a single virus at the place of formation [11]. Therefore aerosol droplets of less than 1 µm diameter in our sampling results have a negligible contribution to transmission of SARS-CoV-2 and were excluded from our work.

Once leaving the oral cavity, dehydration of droplets starts. The smaller the hydrated diameter, the more rapid the shrinking process. The effect is most prominent below a diameter of 100 µm. The shrinking factors suggested, based on experiments, vary from three to five [10,11]. Above 100 µm hydrated diameter, dehydration also occurs, but at a lower speed. Larger droplets are highly unlikely to reach the aerosol fraction, as they settle on surfaces or the floor beforehand [10].

The reported results of our research would reflect the hydrated diameter of the droplets at the time of formation. We obtained the hydrated diameter by multiplying the sampling result of the first sampling head by the shrinking factor established between the two sampling points.

The calculation to obtain the total amount of liquid released has two steps. First, we multiply the total number of droplets per channel (1 to 32 µm diameter) with the volume of a particle in the respective channel. To obtain the total amount of liquid released by an individual, we added the amount of liquid per channel. 

The viral concentrations considered in our analysis range from 10^6^ to 10^11^/mL, based on the work of Wölfel and colleagues, and Jones and colleagues [18,19].

### 2.3. Estimation of Infection Risk

We present the sampling results from a subset of the original study population where sampling data for one hour are available. The one-hour sampling shows different time-periods for simple breathing through the mouth, speaking and speaking with a raised voice. We multiplied the identified time-slots to reach a full hour of breathing, speaking or speaking with a raised voice, each. 

We worked from the assumption that a one-hour meeting is the most frequently observed meeting time during business meetings. During a meeting, people will be quiet, talk or even be excited. 

The exact proportions of each activity over one hour by a person spreading the virus are not known. We propose an empiric and reasonable scenario for the combination of the three activities. We presume that a person shedding the virus during a one hour meeting is breathing quietly for 30 min (50% of the time allocated), is speaking on and off for a total of 27 min (25% of the time allocated) and might talk with a raised voice for about 3 min (5% allocated).

We applied the Wells–Riley equation to estimate the potential risk of infection, similar to Quian and colleagues, and Buonanno and colleagues [20,21]. 

We used the following initial conditions for the Wells–Riley equation:We assume one person spreading disease, entering the room at the same time as those susceptible to the disease.A defined release over one hour of infective particles by the spreader (q), which are completely dispersed in the room.A constant removal of viral particles Q, which, as per Buonanno and colleagues, is a composite factor (addition) comprised of the air-change rate, the settling time of droplets and the proportion of virus dying over time [20].We presume people, beyond the person spreading the virus, being present in the room are fully susceptible to infection.The number of people in a room and the volume of the rooms used in the formula is based on the evaluations and recommendations of the pandemic planning team of Novartis of March 2020. People must adhere to a distance of 2 m during a meeting without additional protection. The 2 m distance correlate to 4 m^2^ area per person. Two people would need 8 m^2^ surface area at a minimum. The average ceiling height of the meeting rooms equals 3 m (personal information from the Head of Facility Engineering at Novartis). Together with ancillary surface areas, we estimate a meeting room for two people to have 30 m^3^. Consequently, we calculated the room volumes for meetings of four and six people in the same manner to get the applied 60 m^3^ and 75 m^3^.No control at source (face-mask) is in use.

We used the infective quanta per hour, q, as derived from the amount of liquid released. A potential risk of infection above 0.05 (5%) in the calculation of the Well–Riley equation is considered unacceptable.

## 3. Results

### 3.1. Sampling Results

As an example, we show the CO_2_ and aerosol sampling in one person (Figure 2 and Figure 3). The curves of the other three participants had the identical picture. This finding indicates that the sampling head of the Grimm device captures a sufficient proportion of droplets from the exhaled air. 

The shrinking factor of 1.6 of the droplets sampled in our set up was measured experimentally with the data of two spectrometers at 25 and 50 cm distance (see also Figure 1). We applied this value on the raw data to obtain the hydrated volume of liquid reported in Table 2.

Figure 2 and Figure 3 show the detailed results for the same person, either as full time-course (Figure 2) or type of activity (Figure 3).

In Figure 2, the blue line, the scale on the left, represents the CO_2_ release over time in one subject. The distribution of aerosols over time is given by the red line, the scale on the right (capped at 50 nL/m^3^).The time scale is on the ordinate.

In Figure 3, the blue line indicates the CO_2_ levels over time, the scale on the left. The red line signifies the concentration of droplets (nanoliter/m^3^) over time, the scale on the right. The letters indicate the activity of the person. The pure data sampling time with the test person is 45 min.

All four adults completed a full hour session. Results for speaking with a raised voice are only available for two actors using the stage.

Of note are several peaks of aerosol release up to five meters from the sampling head. This pattern typically appeared when people were really shouting. Due to increased distance to the sampling head, we did not integrate our sub-set analysis. 

Figure 2 and Figure 3 are the best examples out of the 48 originally sampled people.

Table 3 summarizes the concentrations of liquid released by activity.

To obtain the total volume of liquid released per hour, we multiplied the mean concentration in nanoliters by the respiratory volume of one hour of 0.72 m^3^ for breathing quietly, and 1.375 m^3^ for speaking and speaking with a raised voice [22,23]. The total amount of liquid released, summarized in Table 4, increases with the type of activity, as expected.

### 3.2. Estimation of Viral Load in Liquid Released

The volume of an aerosol droplet of 1 µm^3^ volume equals 1 fL. Such a droplet has a diameter of 1.26 µm. In turn, a droplet of 1 nL (or 10^6^ fF) has a diameter of 126 µm. In contrast, a droplet of 1 µm diameter has a volume of 0.524 µm^3^ (0.524 fL). These relationships are important for the understanding of how much virus may be in which fraction of droplets released by a person.

In multiplying the measured liquid volumes from table four with increasing concentrations of virus per nanoliter and applying the estimated Human Infective Dose 50 of 280 viral particles from SARS, we get the amount of quanta (q) per hour [24].

Table 5 summarizes the calculations of the different levels of quanta.

The column titled “Combination” reflects our assumption of how much time a person might be speaking during a one-hour meeting: 30 min breathing (being quiet), 27 min cumulatively of talking and 3 min of talking excitedly.

### 3.3. Probability of Infection

In applying the Wells–Riley equation, we can estimate the probability of infection of an aerosol-transmitted disease for a specific time in a specified volume of a room. 

The Wells–Riley equation is written as follows:(1)$$Pinf=1-e\;\exp\left(-\frac{Iqpt}{Q}\right)$$
where

*Pinf* = Probability of infection

*e =* Euler number

*I* = number of person(s) spreading disease

*q* = infective quanta per hour

*p* = Ventilation rate of susceptible person(s) in the room in m^3^/hour

*t* = time expired in hours

*Q* = Removal rate of infective agent in m^3^/hour; (sum of air changes, settling time, and dying of virus; multiplied by volume of the room)

We applied the following numbers in Equation (1):

*I* = 1: The probability of a single asymptomatic spreader being present is set to 100%, knowing that the true probability is contingent upon disease activity in a given area/region and hence substantially less.

*q* = Infective quanta released per hour, value taken from Table 4.

Watanabe and colleagues’ estimation is the most plausible approximation of how many viral particles are needed to infect 50% of those exposed to SARS-CoV-2 [24]. SARS and SARS-CoV-2 share the same mode of entry into cells, with SARS-CoV-2 having the possibility to infect cells in the upper respiratory tract more easily [18].

*p* = 0.72 m^3^/h.

This value is the amount of air breathed per hour of a susceptible individual used in the calculation. Office work is considered “light exercise”. Based on the calculations by Zuurbier and colleagues, we use 12 Liters per minute or 0.72 m cube per hour [22]. 

We multiply the respiratory ventilation rate by the number of susceptible people, as established by Riley and Wells (as used by Quian and colleagues, and Buonanno and colleagues in their risk assessments) [19,20]. We use factors of 1, 3 and 5 to account for meetings with 2, 4 and 6 people.

*t* = time spent in the same room in hours (spreader and susceptible persons). In our calculation, we use 1 h.

*Q* = Removal of virus in m^3^/h. This factor represents the reduction and dilution rate of infective particles. “*Q*” is the sum of air-changes per hour, the settling time of particles per hour and dying of virus per hour, multiplied by the room volume. 

We use room volumes of 30, 60 and 75 m^3^, as explained above. The air changes are set at 2 per hour, as we assume the rooms are centrally ventilated. This air-change rate is the most frequent setting in many buildings of Novartis (personal information from the Head of Facility Engineering at Novartis). The settling time is a constant of 1.44/h. It is based on the settling time estimated by Stadtnytskyi and colleagues of 0.0006 m/s for their particles sampled, and a height of 1.5 m. The constant dying time of the virus is 0.63/h, based on the half-life of SARS-CoV-2 in air of 1.1 h, as described by Buonanno and colleagues [11,25,26].

Depending on the flow of air while breathing or speaking, a person releases increased amounts of fluid in the aerosol fraction over time. The virus concentration in the oral fluid in turn defines how “infectious” the aerosol fraction of an individual may be.

Speaking with a raised voice for one hour will raise the quanta (q) above 1 at concentrations between 10 and 100 viral particles per nanoliter (10^7^ to 10^8^ per mL). Breathing normally, speaking with a normal voice and the combination of breathing, speaking normally and with a raised voice for one hour needs viral concentrations of 100 to 1000 per nanoliter (10^8^ to 10^9^ per mL) for increasing the quanta (q) above the level of 1.

Even if people observed a 2 m distance in a meeting room of the dimensions applied, the probability of infection could increase above 5%. This probability is a function of the amount of fresh air added, the time spent together and increasing viral concentrations in the exhaled fluid. 

Table 6 summarizes the thresholds of 5% infection probability for the different meeting situations analyzed.

## 4. Discussion

People with no, little, or unspecific symptoms may spread SARS-CoV-2 unnoticed. Additionally, there is considerable variability of liquid released by individuals. Additionally, people might shed virus at different levels, orders of magnitude apart as shown by Wölfel and colleagues [18,19]. With increasing numbers of people positive in tests, anyone could be spreading the virus.

Sampling of air using liquid impinger biosampling devices at 1.50 m height and a sampling time of three hours returned two culture positive samples out of 14 in a hospital ward with seriously ill COVID-19 patients. The positive samples were both in a corner close to a patient bed. The result is consistent with potential airborne transmission of SARS-CoV-2 [27].

Our data demonstrate that aerosol-based transmission of SARS-CoV-2 in a one-hour meeting needs elevated virus concentrations in the fluid of the mucous membranes of the upper respiratory tract of a person spreading the virus. The risk of infection increases with the loudness of the voice, driven by an increase of liquid released. Asadi and colleagues showed that the amount of particles released increases with the vocalization amplitude [13]. Their study confirms our results in general. The major difference to our study is the fact that Asadi and colleagues use the particle count while reading a text with different vocalization [13]. Our results are based on the transformation of particle counts by diameter fraction to the total liquid volume released by an individual. 

We noted detectable aerosol peaks at increased distances from the sampling point, especially when people were shouting. Our sampling set-up cannot explain these results. We cannot rule out bias from ordinary dust particles from a variety of sources contaminating the results. If the peaks were remnants of the original droplet cloud, additional research would be necessary to support any such hypothesis.

We restricted our analysis to ventilated small meeting rooms, people present in the room maintaining a physical distance of 2 m and one person spreading the virus, but being asymptomatic. We consider this scenario as most appropriate for meetings in many circumstances.

Müller and colleagues propose a different use of the Wells–Riley equation for the risk assessment of transmission of SARS-CoV-2. They used the total aerosols exhaled over time and the probability of a “spreader” being in the room, based on actual epidemiological data. Müller and colleagues elegantly enlarged the possible scenarios for risk assessment by dividing the absolute risk of infection of a given situation by the one of a reference situation. The result of their work is a set of graphs indicating above which number of people, air-change rate in the object and the risk in the reference object would lead to an overall increased risk of infection [28]. 

Our data directly inform the amount of aerosol released and could be used in this type of risk assessment.

Jones and colleagues also proposed a matrix for risk of infection. Their elements are occupancy of rooms, meeting time, activity (breathing, speaking, and shouting), and wearing of face coverings. The authors clearly demonstrated that the risk of infection increases with occupancy and presence of SARS-CoV-2 in most situations reviewed [29].

Our data are consistent with the results of this risk-assessment matrix for meetings without the use of face coverings

Miller and colleagues did an in-depth review of the Skagit Superspreading Event (SSE), where 32 to 57 (53 to 87%) of the 61 participants of a 2.5 h choir practice developed COVID-19 after the practice on March 10, 2020. The main mode of transmission during the SSE is considered to be inhalation of droplet aerosols. The important factors established were closed rooms with ventilation running, long hours of being together in a single room and the presence of a single person spreading substantial amounts of virus [30].

Our data are consistent with the results of Miller and colleagues, as we can demonstrate that room volume, occupancy, ventilation rate and time spent together are the defining factors increasing the probability of infection with SARS-CoV-2.

In another seminal major literature review, Cirrincione and colleagues proposed a protocol intended to support the suppression of spread of SARS-CoV-2 at work. Additionally, they extensively reviewed the strategies applied to fight the virus. The authors proposed a risk classification scheme supporting the implementation of workplace programs, based on documents mandatory in Italy or the US-OSHA. The authors proposed a well-founded set of organizational, environmental, and personal control measures, linked to the risk-level established using their risk assessment approach.

The document of Cirrincione and colleagues is most useful for workplaces in the health care industry. Important points from this document can also be extracted for use in other industries with different work-situations. Namely the reduction of contacts (limit people at work, establish work from home policies), use of face masks, when meeting people, increased personal hygiene and cleaning, and having a sneeze/cough etiquette implemented are the most relevant elements for workplaces in general.

Our data support the implementation of the organizational, environmental and personal aspects of the review of Cirrincione and colleagues in workplaces in general [31]. 

Under the assumption of both droplet and aerosol transmission contributing different proportions to transmission of SARS-CoV-2, reducing droplets at the source is paramount in prevention of transmission. 

Capturing droplets by face-masks sufficiently retaining droplets and aerosols is highly effective and therefore a key measure of prevention of transmission of SARS-CoV-2 [32,33,34,35].

A major limitation of our work is the concern of underestimating the amount of liquid released while breathing or speaking. Despite being an open sampling set-up, the CO_2_-curve parallels the aerosol count, as shown in Figure 2 and Figure 3. 

There is a clear correlation of aerosol count and liquid released by an individual. Droplets shrink once they leave the oral cavity. Droplets of 6 to 32 µm diameter will shrink to less than 5 µm diameter rapidly. We considered whether our sampling results, using respirable aerosols, could potentially reach the periphery of the lungs. Droplet aerosols float and tend to settle very slowly [10]. Air-direction from fans or turbulence from any source may therefore propel the aerosols into the breathing zone of susceptible people. 

Another potential source of loss of particles in our setting could be the effect of the thermal plume. 

Feng and colleagues studied the flow of aerosols of 0.3 to 1.0 µm under the effect of the thermal plume for a single manikin sitting, standing or lying in bed. Sitting had the greatest plume-effect, especially when the room temperature was substantially lower than the temperature of the body. Particles at the size studied by Feng and colleagues rarely penetrate the thermal plume [36]. We considered the effect of the thermal plume to be minimal on our results and interpretations. The droplets potentially containing SARS-CoV-2 used for our analysis are large enough to penetrate the thermal plume, as we sampled them at a 25 cm distance. Exhaled droplets of ≤1.0 µm diameter at the exit from the nose or mouth are highly unlikely to contain any viral particles. While numerous in numbers, the individual volume of a droplet of ≤1.0 µm diameter leads to a negligible probability of containing a single virus, even at very high concentrations at the location of formation. Stadtnytskyi and colleagues considered droplet sizes below 3 µm at the time of their formation too small to contain a single virus [11]. From our review of viral loads and droplet volumes, we concur with this assessment. 

Our conclusion from these facts is that our sampling results are representative of the overall exhaled aerosols by asymptomatic people potentially spreading SARS-CoV-2.

Another limitation could be the different amounts of liquid released. Our results are lower than the ones extrapolating from experiments with substantially shorter sampling times. Our set-up reflects the reality of aerosol release over one hour better, as we captured different bursts of aerosol release.

The use of the Wells–Riley equation could be an additional limitation of our work. The original assumption of instantaneous dissemination of the virus into the room is difficult to maintain, especially in larger rooms. While we agree to this fact, restricting our analysis to rooms of not more than 75 m^2^ and applying a meeting time of one hour allows, in our view, a risk analysis based on this equation. 

We acknowledge that other combinations of the time for breathing, speaking and speaking with a raised voice to the one applied by us would change the amount of quanta released. However, the effect of a different proportion will not change the overall outcome of our interpretation that meetings with two people or more in a small room and for at least one hour may lead to infection of naïve persons. 

Our work is relevant for the workplace setting with few people meeting in a small meeting room. It supports the conclusions of Vella and colleagues and the protocol of Cirrincione and colleagues [4,31]. It would also support a risk assessment as proposed by Müller and colleagues [28]. 

Our work supports implementation of preventative measures to curb the spread of SARS-CoV-2. We suggest focusing prevention of SARS-CoV-2 more on capturing droplets at the source. This effect can be enhanced by combining lower room occupancy, short meeting times, excellent hygiene and physical distancing. Our analysis supports recommendations to increase the air-changes of the ventilation system or frequently opening windows to this effect. Stagnant air or poor ventilation will allow the accumulation of droplet aerosols in a room. People using such a room later in back-to back meetings may then be infected.

## 5. Conclusions

It is impossible to predict who is shedding virus and to what extent, especially for asymptomatic people.

Increasing incidence rates of COVID-19 across the world and the upcoming winter on the northern hemisphere mandate meticulous implementation of generic preventative protection measures by individuals and policy makers alike. The measures should include early use of facemasks in indoor environments in addition to distancing, hygiene and sneeze/cough etiquettes. The goal must be prevention of spread of SARS-CoV-2 by substantially reducing the amount of droplets released, so reduction of contacts are not practicable.

The observed detection of aerosol peaks at distance in our sampling warrants further research, as they might be important in assessing the risk of transmission of SARS-CoV-2.

## Figures and Tables

**Figure 1 ijerph-17-09088-f001:**
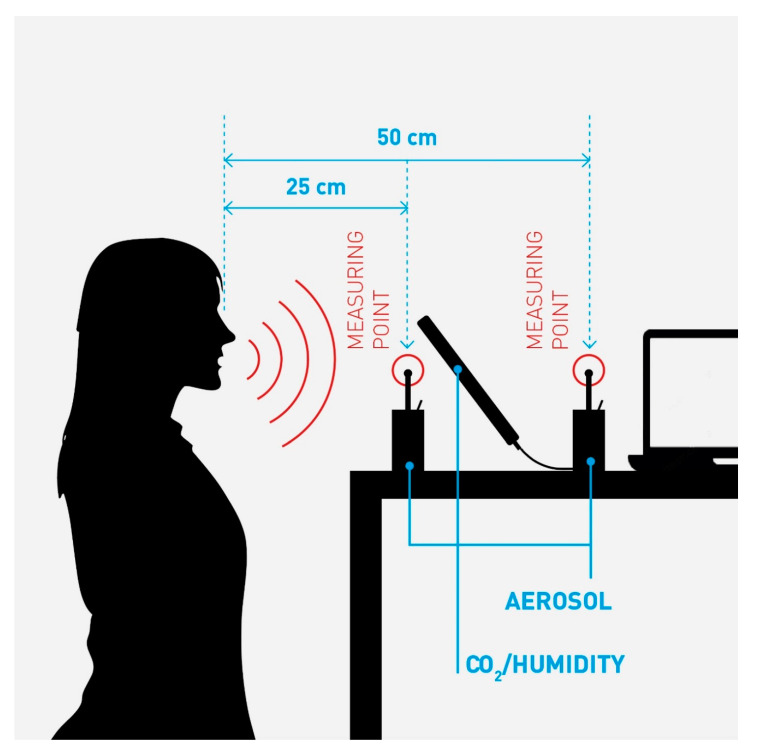
Schematic picture of the sampling set up.

**Figure 2 ijerph-17-09088-f002:**
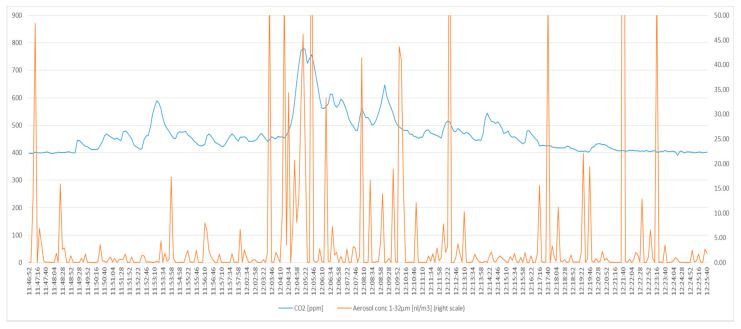
Distribution of aerosols and CO_2_ over time.

**Figure 3 ijerph-17-09088-f003:**
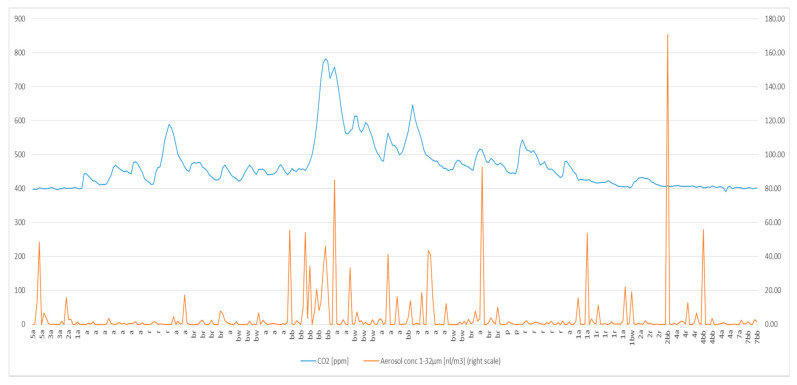
Distribution of aerosols and CO_2_ over time indicating the current activity (see Table 2) as ordinate.

**Table 1 ijerph-17-09088-t001:** Main factors for transmission of SARS-CoV-2.

Transmission	Spreader	Susceptible Person
Favored	Viral load	Proximity, lack of hygiene
Favored	High breathing volume	High breathing volume
Favored	Small room volume, low Ventilation rate	Time spent together in room
Favored	Sneezing/Coughing	Proximity, lack of hygiene
Prevented	Face mask: sufficient retention	Face mask: sufficient retention
Prevented	Good hygiene	Good hygiene
Prevented	Sneeze and Cough etiquette	Sneeze and Cough etiquette
Prevented	Self-isolation with positive test, regardless of symptoms	Not applicable

**Table 2 ijerph-17-09088-t002:** Explanation of the letters in Figure 3.

Abbreviation	Activity
A	Breathing
R	Normal voice
Br	Raised voice
Bw	Angry voice
Bb	Shouting
P	Singing
Number	Distance to sampling head in meters

**Table 3 ijerph-17-09088-t003:** Concentration of liquid released over one hour, by activity.

Activity	Sample Size	Mean Concentration	Standard Deviation	Minimum	Maximum
Breathing	4	1.28 nL/m^3^	1.25	0.41	3.44
Speaking	4	1.67 nL/m^3^	1.04	0.54	3.08
Raised Voice	2	4.44 nL/m^3^	2.58	1.86	7.02

**Table 4 ijerph-17-09088-t004:** Amount of liquid released over one hour, by activity.

Activity	Mean Concentration	Breathing Volume	Liquid Volume
Breathing	1.28 nL/m^3^	0.72 m^3^/h	0.92 nL/h
Speaking	1.67 nL/m^3^	1.375 m^3^/h	2.30 nL/h
Raised Voice	4.44 nL/m^3^	1.375 m^3^/h	6.11 nL/h

**Table 5 ijerph-17-09088-t005:** Quanta released (q) as function of the starting virus concentration.

Virus/mL	Virus/nL	Quanta/h Breathing	Quanta/h Speaking	Quanta/h Raised Voice	Combination 30/27/3 Minutes
10^6^	1	0.003	0.008	0.022	0.006
10^7^	10	0.03	0.08	0.22	0.06
10^8^	100	0.3	0.8	2.2	0.6
10^9^	1000	3	8	22	6
10^10^	10,000	33	82	218	63
10^11^	100,000	330	821	2182	640

**Table 6 ijerph-17-09088-t006:** Thresholds of 5% infection probability for different meeting rooms.

Activity	Virus/nL	Quanta/h	People in Meeting	Percent Fresh Air	Room Volume
Breathing	1000/nL	3	6	50%	75 m^3^
Speaking	1000/nL	8	2	100%	30 m^3^
Raised Voice	100/nL	3	6	50%	75 m^3^
Raised Voice	1000/nL	16	2	100%	30 m^3^
Combination	1000/nL	6	2	50%	30 m^3^
Combination	1000/nL	6	4	100%	60 m^3^

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
