# Peer review of "Aerosol Release by Healthy People during Speaking: Possible Contribution to the Transmission of SARS-CoV-2"

_ijerph, 2020, doi:10.3390/ijerph17239088_

Round 1

Reviewer 1 Report

Firstly, I thank the Editorial Committee for the opportunity to review this manuscript. The authors present a relevant study during the current times of pandemic, contributing to a better understanding of the conditions in which aerosol transmission is occurring (following the current WHO call updated 20 October 2020).  Furthermore, the proposed manuscript meets adequately the purposes of the journal. It should be noted the authors do not present an original study, but an excerpt of a data set from a previous study.

However, some recommendations are suggested below in order to improve the quality and methodological rigour of the manuscript for its publication.

In the Introduction section, it should be advisable to include the meaning of “NA” in Table 1. I consider the paragraph from line 50 to 54 should be removed or included at the end of this section, justifying the rationale of the study (removing the name of an author of the manuscript). Please, use “.” and not “·” for decimal numbers. In general, this introduction section is well carried out and referenced).

The Material and Methods section should be improved. In this section is included neither the study setting nor its timing. The authors state they carried out and extrapolation and re-analysis of a data set from a previous study, but they do not describe in-depth this process. I consider the sample is quite scarce (only four participants). I have read the original paper and two participants were opera singers: a tenor and a mezzo-soprano. In this sense, do they produce more droplets than people normally do? In the original paper, the selected participants singed, not spoked with a raised voice. Is it correct? The authors assumed a combination of intervals of breathing, speaking normally and speaking with raised voice (please, show also its percentages), but how they estimated this timing to emulate a one-hour meeting? The decision date of ethical approval is not included. I consider the assumptions for the Wells-Riley equation may be controversial if they are not well justified. Furthermore, I consider the effect of the thermal plume of an individual on the aerosol dispersion in a room and the ventilation systems constitute a study limitation and, consequently, it should be included in the proper section.

In general, the Results section is based on the authors’ assumptions which may be controversial. It should be noted the extrapolation of the results for meeting with 2,4, and 6 people, and the room volumes of 30, 60, and 75 m3.  I recommended all assumptions would be well-justified. In Figure 2, please include the meaning of the letter which indicate the activity of the person. In Table 2, why the sample size in the activity raised voice was only 2?

In the Discussion section, it should be recommended to specify the differences between the present study and other studies which obtained consistent results (mainly the study carried out by Asadi et al., 2019). In line 293, the authors mentioned data not shown in the present study, and I consider they would be shown to be discussed. Regarding study limitations, it should be included (besides the above-mentioned limitation) the fact that aerosol emissions vary significantly from person to person and actors/singers may produce more droplets than people normally do. Furthermore, it could be interesting include the fact aerosol emissions could remain in the room if it is not well-ventilated.

Finally, although 80% of references are updated and correspond to this year, the authors have to review all references, mainly publications year (mainly in ahead of prints) and journal names.

Author Response

Dear reviewer, Thank you valuable input. We have changed the document accordingly.

please note that the lines correspond to the ones of the track-change documents.

  1. “NA” in table 1 changed to “Not Applicable”.
  2. Text from original lines 50 to 54 has been removed. Relevant information from the “old” paragraph has been added in 2.1, lines 108-118.
  3. We formatted the decimal points to “.”.
  4. We have added information on the study setting and timing.
  5. We have better explained and in more detail how we calculated the total amount of liquid released from the number of aerosol particulates measured (lines 176-179)
  6. We acknowledge the small number of people participating. The original study had a different context. It was designed to review the overall protection concept of the Swiss National Organization for Orchestras and Stages. For this, only few individual people were necessary. The additional use in a risk assessment based on the Wells-Riley equation arose in a discussion of the two authors of this article. We identified a need to have a better approximation of how many viruses could actually be released from a person. We also noted, that the respirable fraction of droplets hardly may contain a single virus. Hence, we set out to use data from healthy people over approximately one hour as basis for a comparison of risks. The data sets of the four people used in our analysis were the ones that were most complete, of very good sampling quality and therefore suitable for us.
  7. We have added more information on the demographics of the four people.
    The data for the two people speaking with a raised voice stem from two actors (male/female) reciting a text in stage voice. The two singers never did this exercise.
    From the one-hour sampling, we used time-slots identified as breathing, speaking and fro two, raised voice. We would be able to account for how much liquid was released by singing. But this is outside of the scope of the article.

We additionally noted distance effects with peaks at 2-5 m distance from the sampling head linked to shouting or loud voice. Aerosol particles from other sources (ventilation, clothes, movement of people) could corrupt our results. We could also hypothesize that the aerosol cloud reached sampling point. The set-up of the sampling does not allow for verification of this hypothesis, despite video documentation of the activities. Hence, we only note it as observation with the suggestion of more research.

However, if you would be interested in singing data or other insights, please contact Thomas Eiche, who is the industrial hygienist and has all the data.

  1. Our choice of time-allocation is very subjective from our side. Clearly, there are different allocation patterns possible. There will be people that barely say a word, there will be people speaking a lot and there will be excitement (fun, anger). Speaking and especially speaking at a raised voice will dry out the mouth and is not sustainable for long (unless trained). One of the actors vented his frustration on COVID19 in a staccato speech at a raised voice for 1.5 minutes (90 seconds), released an extrapolated 44 nl for an hour (data not shown, as this sampling was outside the one hour sampling). He maintained that he could not repeat this exercise twice and would need quite some time of rest.
    We discussed the ramifications of time allocation on lines 443-447.
  2. We agree to your concerns about the use of the Wells-Riley equation. In general, this equation makes a highly complex situation explainable in simple terms to management and associates.
    To obviate the fallacy of oversimplification, we restricted our analysis to small meeting rooms, which are deemed very frequent in our internal analysis of meeting rooms in Novartis (information from the Head of Facility Engineering). For our internal analysis (using Stadtnytskyis data for estimating the quanta), we restricted ourselves to the room volumes used in our article. In the discussions with engineers, we decided that room volumes of more than 100 m3 are poorly amenable to a risk assessment using the Wells-Riley equation.
  3. We acknowledge your point on the effects of the thermal plume. We discuss these effects in lines 418-430. Of note is the fact that we sample the droplets at 25 cm from the mouth. The shrinking factor in our set-up is 1.6 (smaller than the factor of 3 of others). Hence, a droplet of 1 um sampled has at the orifice of the mouth a diameter of 1.6 um. As such, there is a very small probability of such a droplet containing a single virus. Additionally, in order to be sampled, the droplet must have passed the thermal plume.
  4. We have explained the calculation of the liquid released in more detail in lines 176-179.
  5. We have elaborated in more details now we calculated the room volumes for 2, 4 or 6 people in lines 207-215.
  6. We added two new figures and an explanation to the letters in figure 3 (table 2)
  7. We have removed the text from the old line 293.
  8. We agree to the fact that individuals may release liquid over a substantial range of nano-liters. Also, the amount of virus in the liquid may span orders of magnitude, as discussed on lines 340-342.
  9. We have added an explanation of why we use only data from two people in table 3.
  10. We added a point on the effects of poor ventilation and stagnant air in lines 455-457.
  11. We have updated all references either with the full data (journal, year, volume, pages) or the date of first publication online found via the DOI link.

Reviewer 2 Report

it is appropriate to describe the physical characteristics of the people involved in the study. For example, it would be advisable to also present data on spirometry for example. it is also advisable to include other data regarding the spread of the virus in the discussion (10.26355/eurrev_202007_22296; 10.7150/ijms.47052; 10.3390/SU12093603)

Author Response

Dear reviewer,

Thank you very much for your helpful comments. We have updated the document accordingly. Please note, that the line numbers  in our refer to the track-change document. 

  1. We have added more demographic information on the four subjects (lines 116-118).
  2. We do not have spirometry data. The original purpose for collecting our data was intended to show how much liquid singers and actors would release while performing their trade. You would expect such data in a physiology study of liquid release by people as primary goal. This is not the case in our study, though.
  3. We have integrated the citations of Vella and colleagues (New Ref 4; Lines 38-41 and 449) and Cirrincione and colleagues (New Ref 36; Lines 389-400 and 449) as these two articles are seminal to our work and in line with several other authors (cited and not cited in our work).

4. The article of Zhao and colleagues (your reference 10.7150/ijms.47052) is a meta-analysis of clinical data. The goal of this study is to compare the epidemiological and clinical features of patients in the ICU versus those not needing ICU treatment. The article is very important for clinicians should be cited in all such review articles or studies. However, the document is not applicable to healthy people (with or without the cited risk factors for severe disease), the group of interest to us. The risk of becoming infected does not depend on the risk factors for severe disease. The article is outside of the scope of our work. Therefore, we decided not to cited the document.

Reviewer 3 Report

In this manuscript, the author try to evaluate the potential SARS-CoV-2 infection risk during one-hour meeting. To achieve this goal, the author firstly experimentally measure the total volume of the droplets one person can produce in one hour (30 minutes breathing, 27 minutes speaking and 3 minutes speaking with a raised voice), and then estimate the resultant viral load. Based on the achieved data, the probability of infection can be achieved using the Wells-Riley equation.

The author is suggested to modify the manuscript according to the following comments:

  1. References should be added in the line 66-69.
  2. In section 2.2, the author describe the instrument used to collect the droplets. However, detailed information should be provided, such as the area of collection.
  3. The author use 30 minutes breathing, 27 minutes speaking and 3 minutes speaking with raised voice as the parameter to study the infection risk. Reference should be added to explain why the author use this parameter. If we change the parameter change, the collected data should also be different. 
  4. The author claim droplet with different size may reach to different places, however, they author only calculate the total volume but did not consider the effect of the size of the droplet to the person expose to the droplets. Correspondent discussions should be added.

Author Response

Dear reviewer, Thank you very much for your excellent comments. We have adapted the document accordingly. Please note that the line numbers in our response correspond to the track-change document.

  1. We have added the reference document nicely summarizing the highly complex situation of deposition of particles emanating from the airways. The most seminal and widely cited document in the field of particle-lung interactions or in textbooks of Industrial Hygiene would be our reference 23 from the International Commission of Radiation Protection. However, the information is not easily found and hence, we opted for citation 10.
  2. We added more details to the methods. The Grimm devices samples in pulses of 6 seconds. As we video-recorded the activity in parallel, each activity could be assigned to the corresponding time-slot of the print-out (see Figure 2 and 3).
    The tight measurement grid of 6 seconds also shows that the aerosol output during breathing, speaking etc. is not simply a constant value. From our point of view, measuring outside of an artificial laboratory environment has increased the reliability of the measurement at the cost of a potentially reduced accuracy. To balance the negative effect of reduced accuracy, longer sampling times are necessary. Of note: the overall sampling time for the 48 people was 860 minutes. We used a fraction of 60 minutes data for breathing, speaking and speaking with a raised voice out of this total for our subset of people. The remainder are data on singing and playing wind instruments.
  3. Our choice of time-allocation is very subjective from our side. Clearly, there are different allocation patterns possible. There will be people that barely say a word, there will be people speaking a lot and there will be excitement (fun, anger). Speaking and especially speaking at a raised voice will dry out the mouth and is not sustainable for long (unless trained). One of the actors vented his frustration on COVID19 in a staccato speech at a raised voice for 1.5 minutes (90 seconds), released an extrapolated 44 nl for an hour (data not shown, as this sampling was outside the one hour sampling). He maintained that he could not repeat this exercise twice and would need quite some time of rest.
    We discussed the ramifications of time allocation on lines 443-447.
  4. The spectrum of particles sampled will be already in the respirable fraction (<5 um diameter) or will reach this fraction in a reasonable amount of time. The physics of droplet shrinking are well summarized in our reference 10 (Zhang et al). Once droplets are in the respirable fraction, they tend to float for long periods and therefore move around a room by air-flow direction, walking, or other sources of turbulence. We have added such a point in our discussion (lines 412-417).

Round 2

Reviewer 1 Report

Firstly, I thank the Editorial Committee for the opportunity to review this manuscript again. I congratulate the authors for their great effort to increase the quality of the proposed manuscript, improving mainly its methodological rigour. They have followed all the recommendations and suggestions proposed by the two reviewers. According to my recommendations, they have clarified perfectly all my questions and doubts, improving the manuscript's reading and facilitating its understanding for the reader. Furthermore, they have clarified some issues in the manuscript, including relevant phrases and sentences. Consequently, they have increased significantly the quality of the manuscript for its acceptance. Therefore, the manuscript could be considered as accepted for its publication.

Reviewer 2 Report

Dear Editor,
the ijerph-1005623 manuscript contains very interesting data even if developed on a small number of samples.
The study is deserving of acceptance for publication, but the authors must highlight the study's limitations.